# Autonomous Design of Photoferroic Ruddlesden-Popper Perovskites for Water Splitting Devices

**DOI:** 10.3390/ma15010309

**Published:** 2022-01-02

**Authors:** Alexandra Craft Ludvigsen, Zhenyun Lan, Ivano E. Castelli

**Affiliations:** Department of Energy Conversion and Storage, Technical University of Denmark, DK-2800 Kgs. Lyngby, Denmark; s184365@student.dtu.dk

**Keywords:** light-harvesting, water splitting, photoferroics, high-throughput screening, Ruddlesden-Popper perovskites

## Abstract

The use of ferroelectric materials for light-harvesting applications is a possible solution for increasing the efficiency of solar cells and photoelectrocatalytic devices. In this work, we establish a fully autonomous computational workflow to identify light-harvesting materials for water splitting devices based on properties such as stability, size of the band gap, position of the band edges, and ferroelectricity. We have applied this workflow to investigate the Ruddlesden-Popper perovskite class and have identified four new compositions, which show a theoretical efficiency above 5%.

## 1. Introduction

The development of novel energy devices is required to meet the challenges of increasing energy demand and dependence on fossil fuels. The conversion of solar energy into electricity, using a photovoltaic (PV) device, or fuels, e.g., hydrogen and oxygen from water [1], by means of a photoelectrochemical (PEC) cell, are among the most promising solutions to achieve a green future. Both of these technologies rely on materials that show high stability, optimal light-harvesting properties, and low electron-hole recombination rates. The maximum theoretical efficiency obtainable from a single photoactive material in a PV cell is ≈33% (Schockley–Queisser limit), which corresponds to a material with a band gap around 1.3 eV, under 1.5 G solar irradiation and including all possible losses [2]. The efficiency is much lower for PEC devices, where the minimum required band gap is above 2 eV to overcome the bare energy to split water (1.23 eV), the reaction overpotentials (≈0.1 and ≈0.4 eV for the hydrogen and oxygen evolution [3]), and the Quasi Fermi-level (≈0.25 eV per band edge) [4]. The maximum theoretical efficiency is thus not larger than 7% [5]. Different solutions have been suggested to increase the PV and PEC efficiencies [6], both at the device level, by using solar concentrators and multi-junctions, and at the material level [7], by discovering novel compounds with supreme properties. Two new classes of materials have shown great potential to improve the solar conversion efficiency. (1) Organometal halide perovskites, where organic molecules are embedded in an inorganic crystal, have superior light absorption properties, high electron-hole mobility, and long lifetime, i.e., low electron-hole recombination rate [8]. However, they also show low stability and contain Pb, which can cause health issues. (2) Ferroelectric semiconductors (photoferroics) have two properties that make them very interesting for a new generation of solar energy conversion materials [9]. On one side, they can generate photovoltages larger than the band gap, and from the other side, they show an intrinsic polarization, which spontaneously separates the electrons and holes without the need of a p−n junction or co-catalysts, in PV and PEC devices, respectively [10,11]. By generating photovoltages larger than the band gap, photoferroic materials would be able to easily provide the driving force (reaction overpotentials) necessary to run the hydrogen and, especially, oxygen evolution reactions. This could allow us to use materials with band gaps smaller than the 2 eV, mentioned above; thus, drastically increasing the efficiency of PEC devices. Moreover, the spontaneous separation of the photogenerated charges could solve some of the issues related to low mobility and high recombination rates, which is often solved with the use of co-catalysts, making the device easier and cheaper to produce. Despite the high potential, this technology is still at its early stages. The literature reports only a few photoferroic perovskite materials useful for PV and PEC devices, such as the oxides BiFeO${}_{3}$ and its derivates [12,13,14], KBiFe${}_{2}$O${}_{5}$ [15], Ba${}_{2}$Bi${}^{3+}$Bi${}^{5+}$O${}_{6}$ [16], and chalcogenides [17], but the optimal materials have not been discovered yet. Moreover, recently, we have shown the polarization of InSnO${}_{2}$N can be switched during the oxygen evolution reaction (OER) to reduce the overpotential and thus increase the sun-to-chemical conversion efficiency [18].

In this work, we establish an autonomous workflow in the framework of Density Functional Theory (DFT) calculations to discover new photoferroic materials for PEC devices. This workflow is based on the calculation of stability, electronic, and ferroelectric properties and then applied to the class of Ruddlesden-Popper oxide and chalcogenide perovskites. Four new photoferroic materials have been identified to absorb at least 5% of the incident photons and be promising for one-photon water splitting applications.

## 2. Autonomous Workflow and Computational Methods

Thanks to methodological improvements [19,20,21] and an increase in the computational power, computational methods have been successfully used to design novel materials with desired functionalities and improved performance. Among others [22], high-throughput approaches and autonomous workflows have been used, in combination with DFT calculations, to design better catalysts [23], batteries [24], novel 2D and 1D materials [25,26,27], and solar energy conversion devices [4,28,29,30].

Starting from the properties of the constituent elements, a good photoferroic material should be formed by abundant, cheap, and non-toxic chemicals. The material should then be stable, absorb a good fraction of the solar spectrum, show good intrinsic polarization, suitable electron-hole mobility, and have good photoelectrocatalytic properties. These properties are calculated thanks to descriptors, which are easy to calculate and, at the same time, provide a good estimation of the quantity under investigation. For example, the stability is calculated using a convex hull analysis. The convex hull is constructed considering all the possible competing phases (constituent atoms, binary, and ternary compounds), taken from the Materials Project database, in which the candidate material can be separated [31]. The heat of formation is calculated as the difference between the DFT total energy of the candidate material and the energy of the convex hull at that particular composition. To include metastability [32], we consider a material thermodynamically stable when its heat of formation is up to 0.1 eV/atom. Furthermore, calculations of the mechanical and dynamic stability could be useful to confirm whether the candidate material could be synthesized or not. The light-harvesting efficiency is often estimated by the size of the band gap or full absorption spectrum and the photocatalytic properties with the position of the band edges [4,10,28,33].

The workflow established here to identify photoferroic materials is shown in Figure 1. All calculations are performed using the GPAW code and the Atomistic Simulation Environment (ASE) [34,35,36]. The workflow is implemented in the framework of MyQueue [37]. After having selected an appropriate chemical space, we use a structure prototype approach, in which all possible combinations obtained by decorating the prototype with the different chemicals are calculated. We then reduce the possible pool of candidate materials by considering simple structural and chemical rules, such as the sum of the electrons should be even to ensure that no bands are crossing the Fermi level, the sum of the possible oxidation states should be equal to zero to ensure a charge balance in the unit cell, and the size of the A and B-cations [38]. This reduces the original search space to around 30% of it. For these possible combinations, we calculate the relaxed structures (until the forces are below 0.05 eV/Å) of the different prototypes, their energies and band gaps. These calculations are performed in the framework of the Generalized Gradient Approximation (GGA) using PBEsol as the exchange-correlation function [39]. The simulations are performed in the Plane Waves (PW) mode with an energy cutoff of 800 eV and a K-point density equal to 3 Å${}^{-1}$. We then compare the energies of the different prototypes and if the most stable one is non-centrosymmetric, has a convex hull energy below 0.1 eV/atom, and shows a band gap, we calculate its electronic properties, such as band gap, band structure, and density of states, using more accurate methods, in this case using the GLLB-SC exchange-correlation function [40,41,42]. For a better description of the electron density, we use a k-point density of 5 Å${}^{-1}$ and include the spin-orbit coupling (SOC) correction. To be able to evolve oxygen and hydrogen, the band edges need to straddle the redox levels of water. This is estimated using the geometrical average of the Mulliken electronegativities of the constituent atoms [28,43,44]. For a general A${}_{a}$B${}_{b}$X${}_{x}$ compound, the position of the valence and conduction band edges, $E_{\mathrm{VB},\mathrm{CB}}$, are thus given by
(1)$$E_{\mathrm{VB},\mathrm{CB}}=E_{0}+\sqrt[{a+b+x}]{\chi_{A}^{a}\chi_{B}^{b}\chi_{X}^{x}}\pm E_{\mathrm{gap}}/2\;,$$
where $E_{0}$ is the difference between the normal hydrogen electrode (NHE) and the vacuum level ($E_{0}=-4.5$ eV) and $\chi_{I}$ is the electronegativity of the neutral *I* atom in the Mulliken scale. If the band gap is in the visible range, i.e., with a gap between 1.5 and 3 eV and the band edges straddle the redox levels of water, we proceed to calculate the absorption spectrum using Time-Dependent DFT, which gives a more accurate estimation of the light-harvesting efficiency. The theoretical efficiency, η, is calculated as
(2)$$\eta =\frac{1}{n_{\mathrm{tot}}}\int_{{\mathrm{gap}}_{d}}^{\infty }ph_{\mathrm{abs}}\left(E\right)n_{\mathrm{ph}}\left(E\right)dE\;,$$
where $n_{\mathrm{tot}}$ is the total number of photons emitted by the sun at AM1.5, $ph_{\mathrm{abs}}\left(E\right)$ is the photon absorptivity of the material, and $n_{\mathrm{ph}}\left(E\right)$ the number of sun photons at the energy, *E*, in eV. We assume that no absorption takes place below the direct band gap, ${\mathrm{gap}}_{d}$. Here, since we do not consider phonons, which are required to change the momentum in indirect transitions, we assume that no photons are absorbed below the direct gap. This approach is explained in detail elsewhere in the literature and has been used to estimate the light-harvesting properties in perovskites [45]. Moreover, the ferroelectricity/spontaneous polarization using the Berry phase approximation is estimated [10]. We note here that the indirect band gap materials are relevant only if phonons are involved in the absorption process. If that is not the case, e.g., for thin-films, the relevant gap to consider is the direct value. A material is considered a candidate only if it shows a spontaneous polarization and an efficiency of at least 5%.

Although this workflow can be applied to any crystal structure, we use it here to investigate the perovskite family. Perovskite compounds have shown a manifold of properties from efficient light-harvesting and high stability, superconductivity, and photo and ferroelectricity [46].

Moreover, the perovskite structure is able to host almost all elements from the periodic table, which allows for a very wide range of combinations, optimal for a screening project. Conventional cubic perovskites are, however, centrosymmetric so they will not show any polarization. On the other hand, perovskites exist in many different symmetries, such as double and layered, which can show an intrinsic polarization [47].

In this work, we consider the Ruddlesden-Popper (RP) layered perovskite phase, with formula A${}_{3}$B${}_{2}$X${}_{7}$, where A and B are cations (A = Ba, Ca, Mg, or Sr; B = Ge, Hf, Pb, Sn, Ti, or Zr; $r_{A}>r_{B}$ and ${†}_{A}\leq {†}_{B}$, where *r* is the radius and † is the oxidation number) and X is an anion (X = O, S, or Se). An RP is formed by alternately stacked rock-salt layers (AX) and two perovskite-like layers (with formula $ABX_{3}$) along the *c*-axis of the crystal. Therefore, the RP phase shows some similar properties to the cubic perovskites, and, simultaneously, because of the two different chemical environments (rock salt and perovskite-like layers), more unique properties, including a polar structure, which can cause the generation of intrinsic ferroelectric behavior, as it has been recently shown for selected chalcogenide RP [17,48]. We investigate here the five most common RP prototypes, two centrosymmetric (space groups Ccca and I4/mmm, which by definition cannot show a polarization) and three non-centrosymmetric ($Cmc2_{1}$, Pbcn, and $P4_{2}/mnm$), as shown in Figure 1.

## 3. Results

Figure 2 shows the heat of formation and the band gap of all the calculated compositions. A material that shows good stability and band gap in the desired range is indicated in red. Overall, looking at the heat of formation, most of the investigated materials are stable, or at least metastable. We note that, despite having good stabilities, most oxides show very large band gaps (larger than 3 eV), which remove the majority of them from the pool of candidate, as we are considering only materials with a gap in the visible range. The wide band gaps of the oxides are a result of the large electronegativities difference between metals and oxygen [28,49]. Moreover, the very large electronegativity of oxygen has the effect of generating materials with rather deep bands at lower energies compared to the oxygen evolution potential. This, combined with a band gap in the visible range, makes the band edges not well-aligned with the redox levels of water, causing either large energy loss or making the material not suitable for evolving oxygen and hydrogen simultaneously. Sulfides behave differently from oxides. Firstly, almost all of them are more stable in the $Cmc2_{1}$ prototype, which is non-polar and thus allows for spontaneous polarization. Most of the materials have heat of formations below the metastability threshold, except for the compounds formed by Ge and Pb, which are very unstable. Secondly, the band gaps are smaller than the ones of the oxides and are within the visible light range due to the fact that sulfur is less electronegative than oxygen, which also impacts the position of the band edges with respect to the redox levels of water. Selenides seem even more promising than sulfides. The most stable prototypes are non-polar, the heats of formation are more negative than the ones of the sulfides, and the band gaps are smaller.

A total of 25 compositions survive the criteria based on stability and band gap, as well as having a non-centrosymmetric most stable prototype. Out of these, only 19 show a spontaneous polarization, as indicated in Table 1. The materials that show polarization in the Z-direction have the $Cmc2_{1}$ space group, while the ones where the polarization is in the X/Y-direction have the $P4_{2}/mnm$ space group.

The polarization direction becomes important to construct the water-splitting device. The absorption of light should happen in the thickest direction of the material to allow for an increased light-absorption ratio, while the splitting of the photogenerated charges should occur in the thinnest direction, to avoid their recombination. If the polarization is used to enhance the splitting of the charges, it should point along the thinnest direction and the light-absorption in the perpendicular direction. In practice, if the polarization points towards the Z-direction, then the absorption should happen in the XY-plane, and vice versa.

To be considered as candidate materials for water splitting, the band edges of a material should straddle the redox levels of water, which is a condition to allow the evolution of hydrogen and oxygen from water. Figure 3 shows the position of the band edges of these 19 candidate materials. Only 10 of them straddle both the hydrogen and oxygen evolution potentials, while 9 only straddle the hydrogen level. While the former can be used to run an overall (one-photon) water-splitting reaction, the latter can be used in a tandem device (two-photons) to evolve hydrogen [50,51].

The band gap is the most simple descriptor for the light-harvesting efficiency. This, however, does not take into account the kind of transition or its strength. For this reason, we calculate the absorption spectrum and calculate the number of absorbed photons. This procedure and its details have already been used to estimate the light-harvesting efficiency in cubic and layered perovskites [45]. The theoretical photon-absorption efficiencies are calculated in the direction perpendicular to the polarization as the ratio between the number of absorbed photons and total amount of photons from the Sun (AM1.5), and they are plotted as a function of the band gap in Figure 4. Four compositions (Mg${}_{3}$Ti${}_{2}$S${}_{7}$, Ba${}_{3}$Sn${}_{2}$S${}_{7}$, Ba${}_{3}$Ge${}_{2}$Se${}_{7}$, and Mg${}_{3}$Sn${}_{2}$S${}_{7}$) can be used for one-photon water splitting, with a theoretical capacity above 5%. To our knowledge, only Ba${}_{3}$Sn${}_{2}$S${}_{7}$ has been previously synthesized, however, in a different space group [52]. In addition, we have identified a handful of materials, which can be used for the hydrogen evolution in a two-photon water splitting device, with an efficiency well-above 10%. We note that this estimation of the efficiency does not include recombination losses but is still an improvement of the efficiency obtained from the band gap values.

## 4. Conclusions

In this work, we have described an autonomous workflow to identify new photoferroic materials for light-harvesting in a photoelectrochemical water splitting device. Our workflow has been applied to investigate oxide and chalcogenide Ruddlesden-Popper perovskites. Based on descriptors, such as stability, size of the band gap, position of the band edges, absorption spectrum, and ferroelectrocity, we have identified four new compounds that have a theoretical light-harvesting efficiency above 5%. Five other compositions could be used for a two-photon water splitting device, with an efficiency above 10%. Beyond perovskite structures, this workflow can now be used to investigate any crystal structure both using a similar structure prototype approach and investigating known materials, for example, from the Inorganic Crystal Structure Database (ICSD) [53]. 

## Figures and Tables

**Figure 1 materials-15-00309-f001:**
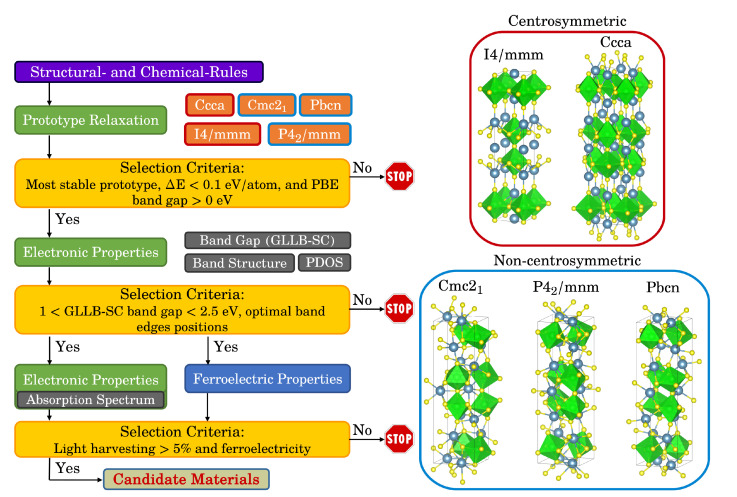
Workflow to autonomously identify photoferroic materials (**left**). The five most common Ruddlesden-Popper prototypes considered in this work (**right**). The A-cation is shown in blue, the B-cation in green, and the X-anion in yellow.

**Figure 2 materials-15-00309-f002:**
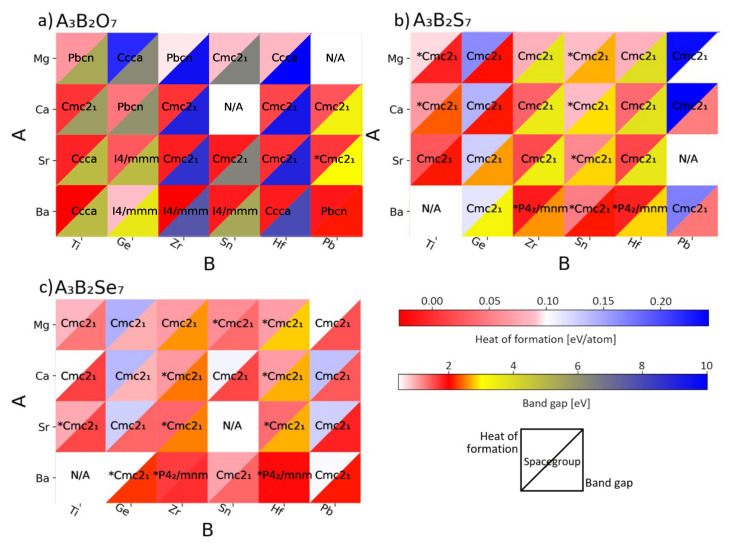
Heat maps showing the heat of formation (top **left** triangle) and band gap (bottom **right** triangle) for the oxide (a), sulfide (b), and selenide-based perovskites (c). A completely red square indicates a stable compound with good electronic properties, which is thereby considered a potential candidate. The space group of the most stable prototype is indicated in each square and stars (*) mark the materials that show an intrinsic polarization.

**Figure 3 materials-15-00309-f003:**
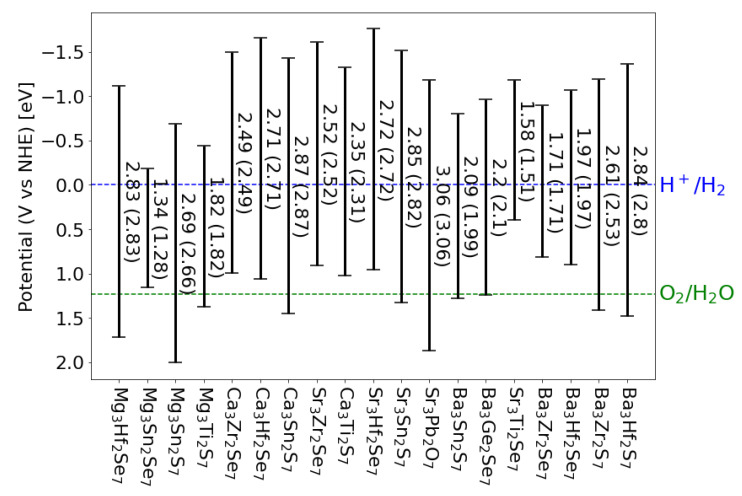
Position of the band edges, calculated for direct gaps, for all the materials that show stability and optimal size of the band gap. The values of the direct (indirect in parentheses) gap is indicated for each composition. The oxygen and hydrogen evolution potentials are also indicated.

**Figure 4 materials-15-00309-f004:**
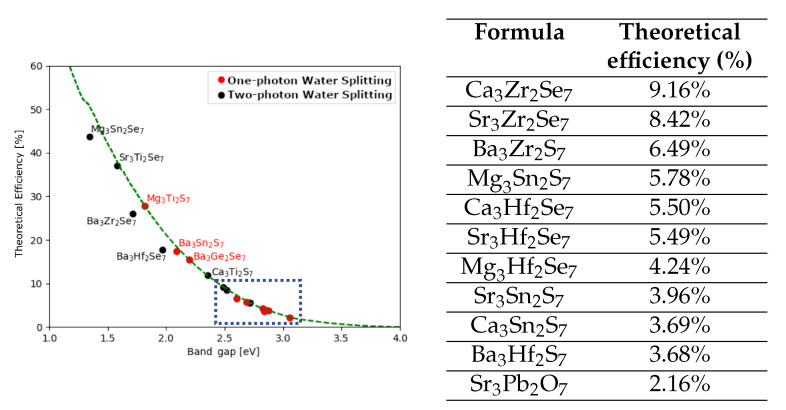
Efficiencies of the 19 candidate materials. The materials indicated in red show potential for one-photon water splitting, while all the others could be used for a two-photon water splitting device. The green line represents the maximum theoretical efficiency. The figure reports materials with efficiency above 10%, while below 10% (enclosed in the dashed box) are summarized in the table.

**Table 1 materials-15-00309-t001:** Calculated spontaneous polarization of the stable candidate materials with a band gap in the visible range. * indicates which materials also have well-positioned band edges, according to Figure 3.

Formula	Pol. ($\mu C/m^{2}$)	Direction
${\mathrm{Mg}}_{3}{\mathrm{Hf}}_{2}{\mathrm{Se}}_{7}$ *	23.97	Z
${\mathrm{Mg}}_{3}{\mathrm{Sn}}_{2}{\mathrm{Se}}_{7}$	46.56	Z
${\mathrm{Mg}}_{3}{\mathrm{Sn}}_{2}S_{7}$ *	31.24	Z
${\mathrm{Mg}}_{3}{\mathrm{Ti}}_{2}S_{7}$ *	51.12	Z
${\mathrm{Ca}}_{3}{\mathrm{Zr}}_{2}{\mathrm{Se}}_{7}$	20.36	Z
${\mathrm{Ca}}_{3}{\mathrm{Hf}}_{2}{\mathrm{Se}}_{7}$	3.72	Z
${\mathrm{Ca}}_{3}{\mathrm{Sn}}_{2}S_{7}$ *	34.29	Z
${\mathrm{Sr}}_{3}{\mathrm{Zr}}_{2}{\mathrm{Se}}_{7}$	8.12	Z
${\mathrm{Ca}}_{3}{\mathrm{Ti}}_{2}S_{7}$	15.72	Z
${\mathrm{Sr}}_{3}{\mathrm{Hf}}_{2}{\mathrm{Se}}_{7}$	24.19	Z
${\mathrm{Sr}}_{3}{\mathrm{Sn}}_{2}S_{7}$ *	8.99	Z
${\mathrm{Sr}}_{3}{\mathrm{Pb}}_{2}O_{7}$ *	31.74	Z
${\mathrm{Ba}}_{3}{\mathrm{Sn}}_{2}S_{7}$ *	10.64	Z
${\mathrm{Ba}}_{3}{\mathrm{Ge}}_{2}{\mathrm{Se}}_{7}$ *	20.58	Z
${\mathrm{Sr}}_{3}{\mathrm{Ti}}_{2}{\mathrm{Se}}_{7}$	24.23	Z
${\mathrm{Ba}}_{3}{\mathrm{Zr}}_{2}{\mathrm{Se}}_{7}$	28.07; 1.12	X;Y
${\mathrm{Ba}}_{3}{\mathrm{Hf}}_{2}{\mathrm{Se}}_{7}$	19.89; 10.16	X;Y
${\mathrm{Ba}}_{3}{\mathrm{Zr}}_{2}S_{7}$ *	24.84; 0.14	X;Y
${\mathrm{Ba}}_{3}{\mathrm{Hf}}_{2}S_{7}$ *	17.51; 7.76	X;Y

## Data Availability

The raw data generated for this work, the workflow for running the calculations, and the script produced to analyse the data are available upon reasonable request and can be obtained contacting I.E.C.

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
