# Peer review of "Autonomous Design of Photoferroic Ruddlesden-Popper Perovskites for Water Splitting Devices"

_materials, 2022, doi:10.3390/ma15010309_

Round 1
Reviewer 1 Report
Remarks to the Author :
Here the authors introduce a well-organized and robust workflow based on some substantial descriptors, including formation energies, band gap size, position of band edges, absorption spectrum, and ferroelectricity, in order to investigate the efficiency of RP perovskites to be used for light harvesting in photoelectrochemical water splitting device. Overall discussion looks like acceptable for common sense and may be useful for particular applications, in this regard I would recommend this work for publication after major corrections.
Herein some major concerns that should be addressed.:
- For clarity, it’s recommended to illustrate the method used for the calculations of band edges. As well, it would be better to include the values of band gap for each system within Figure 3. Additionally, it’s not clear which system shows direct and which one possesses the indirect band gap, such info should be crucial for photoelectrochemical application.
- It’s still not clear for me, the role of ferroelectricity in such application, the discussion about this issue is even poor and insufficient.
- It’s not clear how the authors calculate the theoretical capacity, for instance they stated that four compositions can be used for one-photon water splitting with a theoretical capacity above 5% and 30% for two-photon, how are these value of 5% or that of 30 obtained? Furthermore, the authors reported within the conclusion that the efficiency for a two-photon water splitting device is above 10%, while through the Results (line 182) the value was “above 30%”, why the two numbers are different?
- Figure 4 includes 17 points (red and black), the chemical formula for some structures are reported while others are not, it’s better to report all structures. Moreover, the data reported just beside this figure (including 11 systems) refers to what?
- One grammatical error, line 19, “corresponds to a materials …” should be material or to materials.

Author Response
We thank the reviewer for his/her work. Our reply is attached.

Reviewer 2 Report
The computational work on “Autonomous Design of Photoferroic Ruddlesden-Popper Perovskites for Water Splitting Devices” authored by Ludvigsen et al. is within the scope of the MDPI journal. I recommend this paper for publication after the authors satisfactorily answer my comments below.
- Minor comments: There are numerous grammatical errors in the manuscript. Please correct them in the revised manuscript.
Page 1, line 36 “separate”, line 38 “few”
Page 2, line 55, “increase” and sentence “The material
63 should then be stable, absorb a good fraction of the solar spectrum, …” should be rewritten for clarity.
Page 4, Fig. 2 caption “A completely red square
indicates a stable compound with good electronic properties and thereby a potential candidate”
Page 4, line 132, “We note that despite ….”, line 135, “The wide band gaps of the oxides is a…..” line 143 “Most of the materials have an ….”, and line 164 “To be a considered as a candidate …”
- Major comments:
i) Fig. 1, does not tell anything in crystal structure when there are no labels on atoms. Please label the atom or describe it in the caption.
ii) On page 3, Section 4. Result, can authors provide the expression for the heat of formation energy? Are the formation energies calculated with respect to constituent elements?
iii) I am not fully convinced whether these selected compounds are stable. One cannot confirm the stability of material simply on DFT calculated heat of formation. I am not asking authors to compute dynamic and mechanical stability, although the later calculation would be useful to improve the paper.
iv) Whether or not a compound is suitable for light-harvesting relays on the magnitude of direct bandgap as well as positions of valence band maximum and conduction band minimum with respect to H2O redox potential.
Please, explain in detail how you compute the band alignment. The authors did not say what V vs NHE stands for. Please explain it in text.
v) Can the authors explain the use of the indirect bandgap in Fig. 3? Since no phonons are involved in the photo-absorption and emission process. The authors did not explain it in the main text.
v) Fig 3. is confusing in some cases e.g., Sr3Pb2O7, the only indirect gap is seeable. Perhaps, authors can improve by using a dashed line for indirect bandgap or removing it completely.
vi) Can authors give the expression for absorption phonon?
Author Response

(The authors gave the same response as above.)

Reviewer 3 Report
See attached.

Author Response

(The authors gave the same response as above.)

Round 2
Reviewer 1 Report
The authors have considered all raised comments, and the manuscript has been modified accordingly. The manuscript is currently suitable for publication in Materials.